# Changing the game of time resolved X-ray diffraction on the mechanochemistry playground by downsizing

Giulio I. Lampronti [1,2 ✉], Adam A. L. Michalchuk [3 ✉], Paolo P. Mazzeo [4,5 ✉], Ana M. Belenguer [2,3], Jeremy K. M. Sanders [2], Alessia Bacchi [4,5] & Franziska Emmerling [3]

Time resolved in situ (TRIS) monitoring has revolutionised the study of mechanochemical transformations but has been limited by available data quality. Here we report how a combination of miniaturised grinding jars together with innovations in X-ray powder diffraction data collection and state-of-the-art analysis strategies transform the power of TRIS synchrotron mechanochemical experiments. Accurate phase compositions, comparable to those obtained by ex situ measurements, can be obtained with small sample loadings. Moreover, microstructural parameters (crystal size and microstrain) can be also determined with high confidence. This strategy applies to all chemistries, is readily implemented, and yields high-quality diffraction data even using a low energy synchrotron source. This offers a direct avenue towards the mechanochemical investigation of reactions comprising scarce, expensive, or toxic compounds. Our strategy is applied to model systems, including inorganic, metal-organic, and organic mechanosyntheses, resolves previously misinterpreted mechanisms in mechanochemical syntheses, and promises broad, new directions for mechanochemical research.

[1] Department of Earth Sciences, University of Cambridge, Downing Street, Cambridge CB2 3EQ, UK. [2] Yusuf Hamied Department of Chemistry, University of Cambridge, Lensfield Road, Cambridge CB2 1EW, UK. [3] BAM Federal Institute for Materials Research and Testing, Richard-Willstätter-Straße 11, D-12489 Berlin, Germany. [4] Department of Chemistry, Life Sciences and Environmental Sustainability, University of Parma, Parco Area delle Scienze 17/A, 43124 Parma, Italy. [5] Biopharmanet-TEC, University of Parma, Parco Area delle Scienze 27/A, 43124 Parma, Italy. ✉email: gil21@cam.ac.uk; adam.michalchuk@bam.de; paolopio.mazzeo@unipr.it

Mechanochemistry has emerged as an attractive and sustainable synthetic tool[1,2] with applications for the synthesis of organic[3–7], inorganic[8–11] and hybrid metal-organic molecules and materials[12–15]. It is increasingly clear that many traditional solution-based chemical reactions can be performed in the presence of very little or no solvent using mechanochemical approaches[16]. Providing "greener" and potentially less-expensive strategies than traditional solution methods[1,17–19], mechanochemistry was dubbed by the International Union for Pure and Applied Chemistry (IUPAC) as one of the 10 chemical innovations that will change our world[20]. Despite decades of research[21], the fundamental principles which drive mechanochemical reactions remain poorly understood. This lack in understanding and control represents a significant barrier to realising the full potential of this world-changing technology.

Traditionally, mechanochemical reactions have been studied ex situ, wherein the reaction is stopped and material is removed from the reactor for analysis. Where reaction products are long-lived, ex situ methods provide a powerful means to investigate mechanochemical mechanisms. However, many systems continue to transform even after milling is stopped[22,23]. Moreover, examples are known where the stop-start ex situ approach causes the system to evolve via a pathway alternative to the unperturbed reaction[24,25]. Such reactions can be only studied by directly probing the reaction in situ during mechanical treatment.

Time-resolved in situ (TRIS) monitoring approaches have opened the door to exceptional detail regarding mechanochemical reactions[13]. For example, TRIS Raman spectroscopy[26,27] has allowed unrivalled insight into the rates and mechanisms of ball milling covalent organic chemical reactions and solid–solvent interactions[26], while advances in TRIS X-ray spectroscopy have allowed the study of redox chemistry during nanoparticle mechanosynthesis[28]. As mechanochemistry is predominantly a solid-state synthesis technique, TRIS-X-ray powder diffraction (XRPD) has remained a focal point for mechanochemical investigation. This has included studies on the rates and mechanisms of both chemical[29] and physical transformations[30]. TRIS-XRPD has proved especially promising for identifying unexpected and short-lived intermediates, including indications of short-lived amorphous phases[31]. Despite the significant progress made in the study of mechanochemical reactions through TRIS-XRPD, the quality of attainable data has remained poor. In turn, it has not been yet possible to extract the intricate mechanistic detail required to fully elucidate the mechanisms of mechanochemical transformations from TRIS data.

For instance, the determination of crystallite size which has appeared as a fundamental factor in mechanochemical transformations[32,33], remains largely outside the scope of current TRIS capabilities. Similarly, difficulties with data collection strategies for established TRIS-XRPD methods have demanded the use of large solid loadings, typically hundreds of milligrams[13,27]. However, in many cases, materials are expensive or toxic. Such reactions have not yet been feasible for mechanochemical investigation, even by ex situ approaches which require many individual experiments to map the evolution of a reaction profile. Hence, innovations in TRIS-XRPD approaches to study the mechanism of such reactions promise new directions for solid-state research.

Despite the growing number of TRIS synchrotron studies being reported, the collection of quality diffraction data has proved a difficult task with conventional setups and data collection strategies[13,34,35]. An innovative and highly successful solution was presented by Ban et al.[36,37] who improved the signal to background ratio using a new design of milling jar. In their design, the X-ray beam was focused through a specially designed sampling chamber on the outer circumference of the milling jar,

with small width and narrow walls. Critical to the success of this design is the fast and efficient exchange of powder between the main milling jar body and this sampling chamber. Unfortunately, many materials aggregate into lumps or become sticky, thereby hindering such exchange[30,38]. Most crucially, the jar design does not allow for the addition of liquids. Hence, it is not suitable for common mechanochemical procedures such as liquid-assisted grinding (LAG)[1]. Promising developments for TRIS data acquisition were recently demonstrated for a Resonant Acoustic Mixing (RAM)-induced cocrystallisation[39]. These included the use of a small sample vessel with narrow walls, and the design to reduce the beam path through the milling jar (ca. 0.2 mm from the internal wall) so as to minimise the instrumental contribution to diffraction aberrations such as peak splitting, peak broadening and peak asymmetry (vide infra). This setup is not intrinsically hindered by powder rheology or the addition of liquid. While RAM is inherently different from ball mill grinding as it does not involve ball bearing impacts, the associated developments in diffraction signal optimisation to enhance TRIS data quality have inspired the present work.

We report here an approach for TRIS-XRPD under realistic ball milling conditions combining innovations in milling jar design with data collection strategy, and data analysis. In doing so, we seek to provide solutions to long-standing issues with TRIS-XRPD experiments (Table 1). Our method aims to simultaneously improve both the diffraction peak shape and signal to background ratio whilst providing a readily implementable, practical solution for general use in TRIS-XRPD measurements. We validate and calibrate our TRIS-XRPD approach with a silicon NIST standard[40] and demonstrate its effectiveness with four representative milling reactions, including inorganic, metal-organic and organic mechanosyntheses.

## Results and discussion

We elected to study archetypical examples of reactions from across the major classes of solid materials being regularly studied by ball milling:

Reaction I. inorganic metathesis reaction—the inorganic metathesis reaction $KI + CsCl \rightarrow KCl + CsI$[36];

Reaction II. metal-organic framework synthesis reaction—the formation of a zeolite-like imidazolate framework ZIF-8[13,27,34];

Reaction III. organic cocrystallisation reaction—organic cocrystallisation between theophylline and benzamide to form two polymorphs of the cocrystal; Form I under neat grinding conditions (Reaction IIIa) and Form II under LAG conditions with water (Reaction IIIb)[32,33,41];

Reaction IV. covalent disulfide exchange reaction—the covalent disulfide exchange reaction between bis-2-nitrophenyldisulfide $((2NO_2\text{-PhS})_2)$ and bis-4-chlorophenyldisulfide $((4ClPhS)_2)$ in the presence of a small amount (2 M%) of base catalyst (dbu = 1,8-diazabicyclo[5.4.0]undec-7-ene) to produce the heterodimer 4-chlorophenyl-2-nitrophenyl-disulfide ($2NO_2$-PhS-SPh-4Cl) under neat ball mill grinding conditions[32,42].

By using reported examples, comparison against literature reports of TRIS monitoring from previous setups was possible (Reactions I, II and III). Moreover, our selected systems include high-symmetry crystal structures (Reactions I and II) with strongly scattering elements (Reaction I) and poorly scattering elements (Reaction II), along with low-symmetry systems with poorly scattering elements (Reactions III and IV). Hence, our selection of systems demonstrates unequivocally the promise and universality of our developments.

**Designing milling jars for optimal TRIS X-ray diffraction monitoring.** TRIS-XRPD mechanochemical experiments require

**Table 1 Solutions to long-standing complexities with TRIS-XRPD analysis for ball milling reactions.**

| Complexity | Proposed solution |
|---|---|
| Sample scattering intensity and reliable XRPD refinement | • Low-energy radiation to reduce peak overlap<br>• Minimise jar wall thickness to maximise sample scattering<br>• Include experimental background (empty jar) in the PXRD data analysis by whole-pattern Rietveld refinement to minimise the number of background parameters<br>• Use experimental background scale factor to normalise PXRD scans<br>• Sequential approach to Rietveld refinement, i.e. use the output obtained for scan number "$n$" as input for scan number "$n+1$" |
| Instrumental broadening of diffraction profile | • Low-energy radiation to reduce peak overlap<br>• Optimise beam alignment strategy to resolve multiple scattering components using a standard<br>• Develop physically meaningful XRPD peak shape modelling for microstructural analysis<br>• Include experimental background (empty jar) in the PXRD data analysis by whole-pattern Rietveld refinement to minimise the number of background parameters<br>• Sequential approach to Rietveld refinement<br>• Parametric refinement for phase scale factors, i.e. constrain scale factors to sigmoidal curves |
| Scale of powder required for milling | • Minimise jar wall thickness to maximise sample scattering<br>• Minimise powder caking |
| Loss of free powder by sticking or caking on internal surfaces | • Carefully analyse loading vs milling parameters to maximise powder distribution |

that an X-ray beam pass through the milling jar to the detector (Fig. 1). The photons interact with the jar walls and the powder contained within the jar. The primary aim of TRIS-XRPD measurements is therefore to minimise the scattering of photons from the jar, whilst maximising the signal from the powder. Perspex has been commonly used to make milling jars for TRIS experiments owing to its low absorption coefficient (see Supplementary Fig. 4). However, given the small amount of powder present within the 150 μm X-ray beam at any given time, the diffraction signal of the powder typically remains only weakly visible above the Perspex pattern. With the aim of enhancing the resolution of the attained XRPD profiles, we sought therefore to reduce the X-ray scattering signal from the Perspex wall by designing milling jars with significantly reduced wall thickness. Here, our custombuilt Perspex milling jars comprise walls of 0.5 mm, with internal diameter of 10 mm (see Fig. 1). Additional jars with internal diameter of 12 mm and wall thickness of 0.75 mm were also explored.

**Data collection and Rietveld refinement strategy.** Ideal diffraction occurs from a single point. Conversely, during TRIS-XRPD the beam passes through an elongated sample volume, thereby resulting in broadening, and ultimately splitting, of the diffracted peaks. This technical issue has previously hindered robust analyses of XRPD profiles resulting from TRIS mechanochemical investigations. Additionally, during ball milling, some of the milled powder adheres to the jar walls. This gives rise to the splitting of each measured diffraction peak into three main components (Fig. 1b–d). The inner and outer scattering components arise from powder adhered to the front and back walls of the milling jar, respectively (see $\vec{s_1}$ and $\vec{s_3}$ in Fig. 1). The magnitude of their splitting can be reduced by minimising the X-ray path length through the jar. The scattered intensity between these extremes (see $\vec{s_2}$ in Fig. 1) arises from powder which flows freely within the jar. To model these complex diffraction peak instrumental aberrations, a silicon 640d NIST standard powder[40] was loaded in a jar with one ball bearing and diffraction data were collected during milling: the NIST standard was assumed to have no intrinsic contribution to peak broadening. The triplet peak shape (Fig. 2a) was described with three bell-shaped functions: two split-modified Thompson-Cox-Hastings pseudo-Voigt functions[43,44] (TCHZ, see details in Supplementary Note 5.1)

for $\vec{s_1}$ and $\vec{s_3}$, plus one Gaussian function for $\vec{s_2}$ (see Supplementary Note 5.1 for details). The peak displacement caused by each of the scattering vectors $\vec{s_1}$, $\vec{s_2}$ and $\vec{s_3}$, was corrected by modelling the peak positions with a function reported in Supplementary Notes 5.1 and 8. Moreover, by treating each component independently, a correction for the evolution of sample distribution during milling is intrinsically included.

The exaggerated peak splitting which results from the use of low-energy (17 keV) X-rays proved highly beneficial. By resolving the three scattering components ($\vec{s_1}$, $\vec{s_2}$ and $\vec{s_3}$), high-quality peak profiles could be extracted, evidenced by the impressively low estimated standard deviations for Rietveld refinement of both peak shape and intensity parameters. While the refinement yielded an excellent difference pattern (see Fig. 2a), minor discrepancies in peak position and shape were observed between the experimental and expected silicon standard diffraction profiles at very high scattering angle. We suggest that these distortions result from the non-uniform thickness of the powder adhered to the jar walls. Importantly, these minor profile discrepancies were only visible in the well-defined silicon standard and were not detected in any of the mechanochemical reactions presented herein.

In practice, all samples were collected by aligning the X-ray beam across the shortest chord length through the jar, greatly reducing splitting of the Bragg reflections (Fig. 2b). Careful validation of our Rietveld refinement strategy using the triplepeak model proved that this simplified diffraction pattern can be modelled using a two-component "$\vec{s_1} + \vec{s_3}$" split TCHZ pseudo-Voigt mathematical function (see Supplementary Note 5.1). No signs of scattering from the milling ball were detected in any of the alignment positions presented here. Dedicated studies as to the optimal alignment strategy are ongoing and will be reported elsewhere. For the purpose of this study, Rietveld analyses for all TRIS mechanochemical data were conducted using the diffraction geometry parameters, including the split TCHZ pseudo-Voigt functions, as refined for the silicon standard (see Supplementary Note 5 for details).

For all phases studied in the present work, we have assumed the sample contribution to peak broadening to be related to crystal size only. This approximation is satisfactory in the case of nanosized phases, especially those which do not scatter at high $2\theta$ like most organics. We do not, however, suggest this to be the best

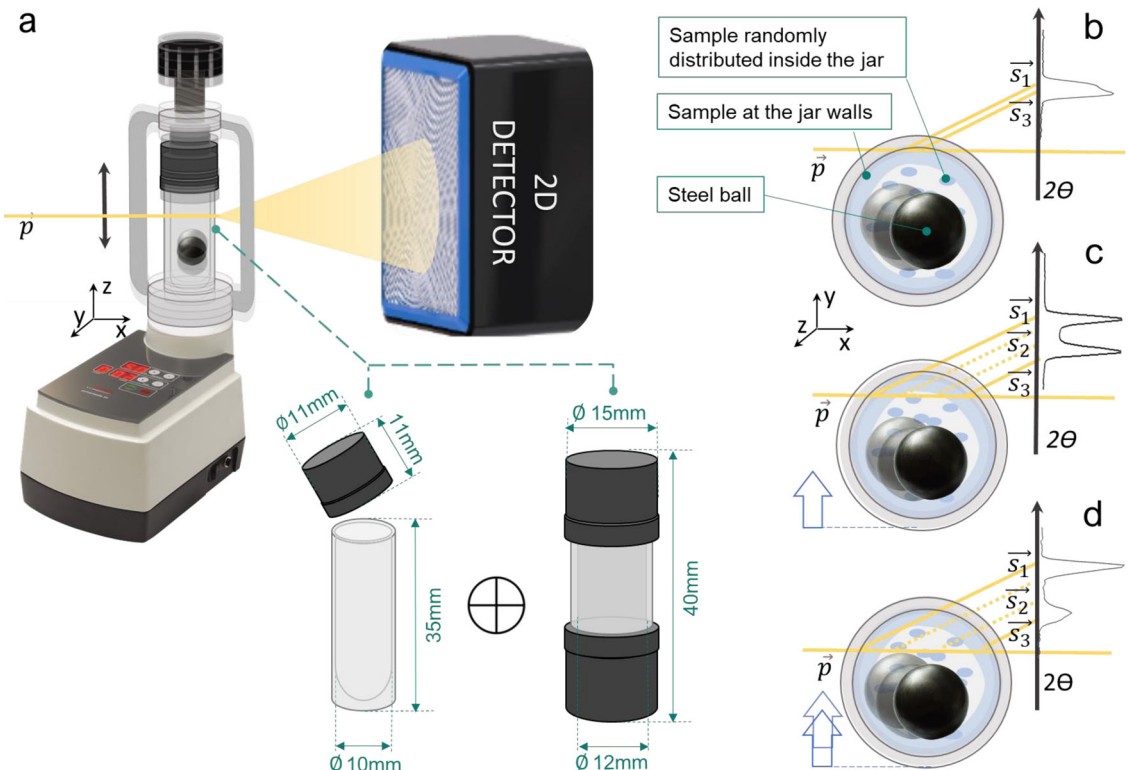

**Fig. 1 Schematic representation of the milling setup used in this study.** The PMMA jars are used with the Fritsch Pulverisette 23 (**a**). The primary X-ray beam $\vec{p}$ (yellow line) passes through the jar and is diffracted by the sample contained within (light blue). Two different jars were developed (see Supplementary Note 2 for details), made from PMMA (transparent) and PVC (black) or stainless steel caps. The jar used for Reactions I–III is 12 mm internal diameter and comprises 0.75 mm walls, whereas the jar used for Reaction IV is 10 mm internal diameter with 0.5 mm walls. Diffraction with this setup results in splitting of each Bragg reflection into a convolution of $2\theta$ positions (**b–d**), as the powder inside the jar is distributed across different locations and hence a range of sample-to-detector distances. Thus, the measured scattering vectors are markedly offset when the jar is investigated in a general position with respect to the primary beam (**c**, **d**) and can be minimised with accurate jar alignment (**b**).

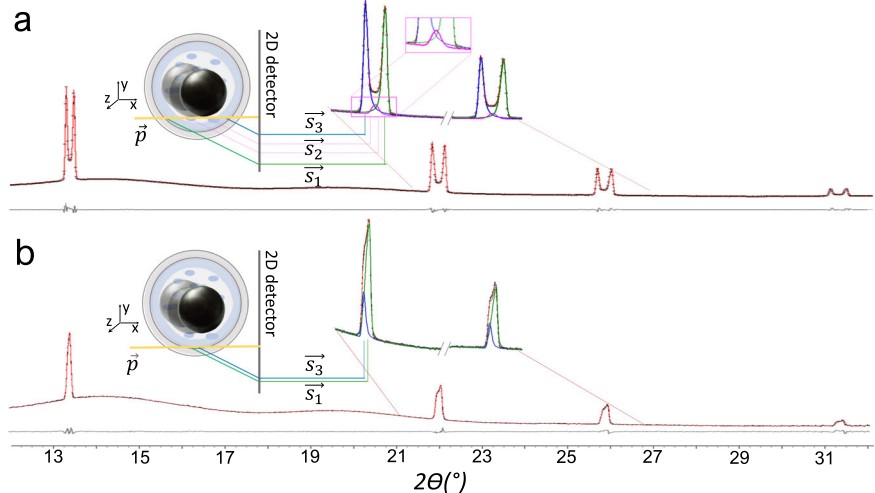

**Fig. 2 Rietveld plot of the Si640d NIST standard at different diffraction geometries.** In each case, the calculated profile is given (red line) against experimental data (black dots) and the difference pattern is shown (grey line). The peak split is compatible with the geometrical constraints typically observed in an in situ mechanochemical experiment. The primary beam $\vec{p}$ (yellow line) passes through the jar and is diffracted by the sample contained within. The contribution to the overall peak shape of the sample distributed within the jar is highlighted in the insets with different colours: the scattering vectors are produced by the sample located at the jar wall closer to the source ($\vec{s_1}$, green line), the wall nearer the detector ($\vec{s_3}$, blue line), and by the sample distributed randomly within the jar ($\vec{s_2}$, pink line). The difference in $2\theta$ angle between scattering vectors $\vec{s_1}$ and $\vec{s_3}$ is larger when the jar is in a general position with respect to the primary beam $\vec{p}$ (**a**). The difference in $2\theta$ angle is minimised when the jar is accurately aligned (**b**), with negligible scattering contribution from the sample distributed within the jar (i.e. $\vec{s_2}$).

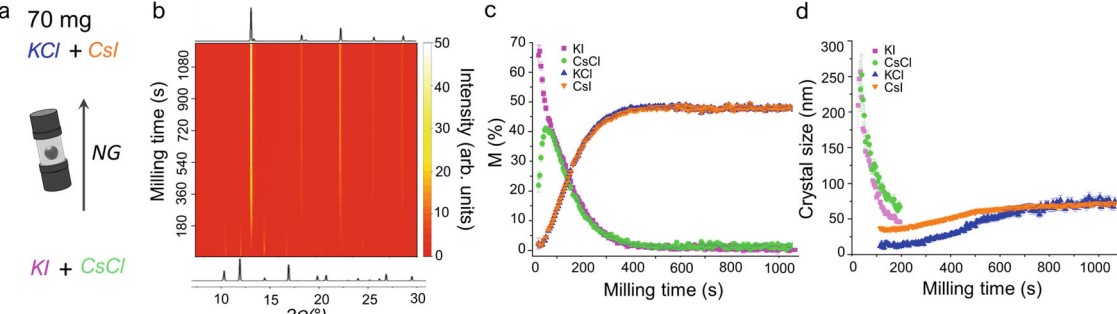

**Fig. 3 TRIS-XRPD for Reaction I under ball mill neat grinding (NG) conditions with a loading of 70 mg. a** The reaction scheme for Reaction I under NG conditions. **b** Heat maps (linear intensity, arb. units) of TRIS-XRPD with empirical background subtracted data for Reaction I, alongside calculated 1D XRPD patterns for the reagents (KI + CsCl) and the final products (KCl + CsI). **c** Quantitative phase analysis for Reaction I obtained by Rietveld refinement, showing the consumption of KI (pink) and CsCl (green), with simultaneous formation of KCl (blue) and CsI (orange) with relative ESD bars. **d** Scherrer crystal size obtained for KI (pink), CsCl (green), KCl (blue) and CsI (orange) with relative ESD bars are shown for phases more abundant than 20 M%.

approach for the general case. On the contrary, we encourage researchers to challenge and improve the strategies introduced here, including microstructural analyses that account for isotropic and anisotropic size and microstrain. An alternative microstructural size and strain data analysis for Reaction I is reported in Supplementary Note 5.7.

**Reaction I: TRIS monitoring of inorganic solid-state mechanochemistry.** We demonstrate first the strength of our TRIS-XRPD approach using the model inorganic metathesis reaction KI + CsCl → KCl + CsI (Reaction I, Fig. 3). For a sample loading of only 70 mg, our approach allowed us to collect high-quality TRIS-XRPD profiles within 5 s. This temporal resolution is comparable with previously reported TRIS-XRPD experiments at significantly higher loadings[36]. Quantitative phase analysis (QPA) performed via Rietveld refinement (see Supplementary Note 5.2) shows the rapid consumption of both reagents (KCl and CsI), with the simultaneous formation of both product phases (KI and CsCl); no significant induction period is visible. Exceptionally precise phase compositions were attainable using our improved data-processing strategy, evidenced by estimated standard errors (ESD) smaller than the size of the symbols themselves in Fig. 3c. Moreover, crystallite sizes with remarkably small ESDs (Fig. 3d) were also obtained for all four crystalline phases (see Supplementary Note 5.2 for details). Thus, in addition to the kinetics of the metathesis reaction, we observe unambiguously the initial comminution of the reagent phases and subsequent growth of the products to ca. 75 nm. The simultaneous comminution and formation of reaction product suggests that for this metathesis reaction, bulk comminution is not a prerequisite for the observed mechanochemical reaction.

**Reaction II: TRIS monitoring of metal-organic framework mechanochemistry.** Next, we explored the mechanosynthesis of the prototypical metal-organic framework ZIF-8 (Reaction II, Fig. 4), which has served as a model system for TRIS-XRPD measurements in various milling setups[13,27,31]. Consistent with previous reports, we could not perform a satisfactory Rietveld analysis of diffraction data for this system. It has been suggested that varying the distribution of guest molecules in the crystallographic model could rectify this fitting[13]. If this were the case, a Pawley refinement would give a perfect fit, wherein the individual peak intensities are free to vary independently and are not constrained by a structural model. However, the subtle—albeit evident—discrepancies between our high-quality experimental data collected using our TRIS setup and the pattern calculated from the Pawley refinement are conclusive evidence of a previously

undetected lattice distortion under milling conditions. This distortion is consistent with observations by TRIS Raman spectroscopic data which, using other TRIS-XRPD setups, remained undetectable[27]. We are currently investigating the exact nature of this structural distortion.

**Reactions III and IV: TRIS monitoring of organic solid-state mechanochemistry.** Historically, the analysis of crystallite size has proved to be a significant challenge for TRIS-XRPD, particularly for poorly diffracting molecular solids. Yet, identifying and monitoring crystallite size evolution is critical for elucidating the intricate mechanisms which drive mechanochemical reactions[32,42]. We demonstrate here how our approach to TRIS-XRPD provides a solution to this critical analysis, using as a model system the mechanosynthesis of two polymorphic forms of the 1:1 cocrystal of theophylline (tp) and benzamide (ba), Reaction III (Fig. 5). Moreover, this analysis can be achieved using a sample loading of 60 mg, whilst providing better-quality data than previously published by TRIS methods at significantly higher loadings[41].

In contrast to the inorganic metathesis reaction (Fig. 3), the neat mechanosynthesis of tp:ba Form I (Reaction IIIa) involves a marked induction period (ca. 500 s). Although no synthesis occurs during this period, there are clear indications of significant comminution of tp crystallites during this induction stage. Notably, signs of cocrystallisation begin only after tp crystallites fall ca. <100 nm in size (see Supplementary Note 5.3 for details on the microstructural analysis). This may be indicative of a critical destabilising size for tp crystallites warranting dedicated investigation. In stark contrast, no induction period is observed in the LAG mechanosynthesis of tp:ba Form II using a trace of water (Reaction IIIb). Rather than comminution, our high-resolution TRIS-XPRD data indicate the previously unobservable, transient formation of tp monohydrate within the first 20 s of ball milling. Our crystallite size analysis again suggests this phase achieves sizes <100 nm prior to visible growth of the product phase, tp:ba Form II. We additionally identify statistically meaningful differences in the equilibrium crystallite sizes of Form I and Form II (see Supplementary Notes 5.3 and 5.4) generated under ball milling conditions. These sizes are consistent with previous ex situ experiments[33], and suggest that each polymorphic form has a unique comminution size limit.

In addition to monitoring crystallite size evolution, a primary aim of our developments was to facilitate the miniaturisation of ball mill grinding technology. Correspondingly, we sought to employ our TRIS-XRPD approach to monitor a remarkably small sample loading: 10 mg. As a model system, we selected the

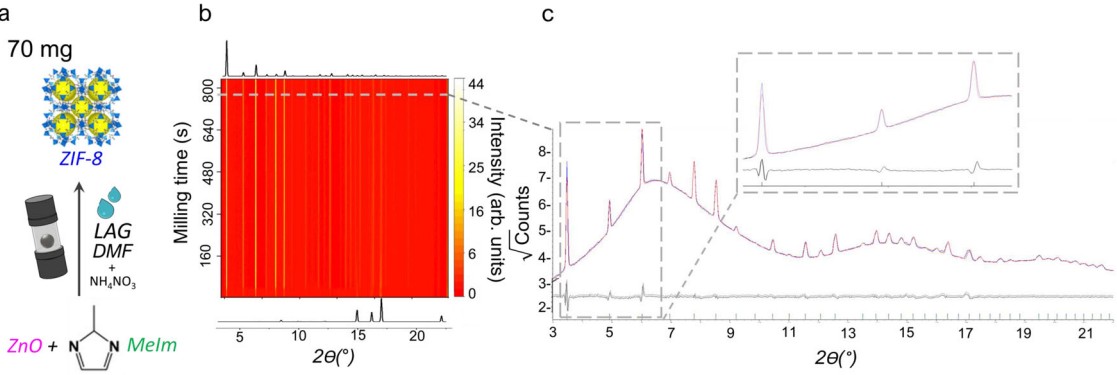

**Fig. 4 TRIS-XRPD for Reaction II, the mechanochemical syntheses of ZIF-8 with a loading of 70 mg. a** A 0.1 equivalent of $NH_4NO_3$ is added to 1 equivalent of ZnO and 2 equivalents of 2-methylimidazole (MeIm) under ball mill liquid-assisted grinding (LAG) conditions with the addition of 17 µl DMF. **b** Heat maps (linear intensity, arb. units) of TRIS-XRPD with empirical background-subtracted data for Reaction II, alongside calculated 1D XRPD patterns for the reagents and the final product (ZIF-8). **c** Pawley fit of ZIF-8: calculated profile (red line) against experimental data (blue line) and relative difference pattern trace (grey line). Inset highlights the discrepancy between the model and experimental data in the lower angle part of the diffractogram.

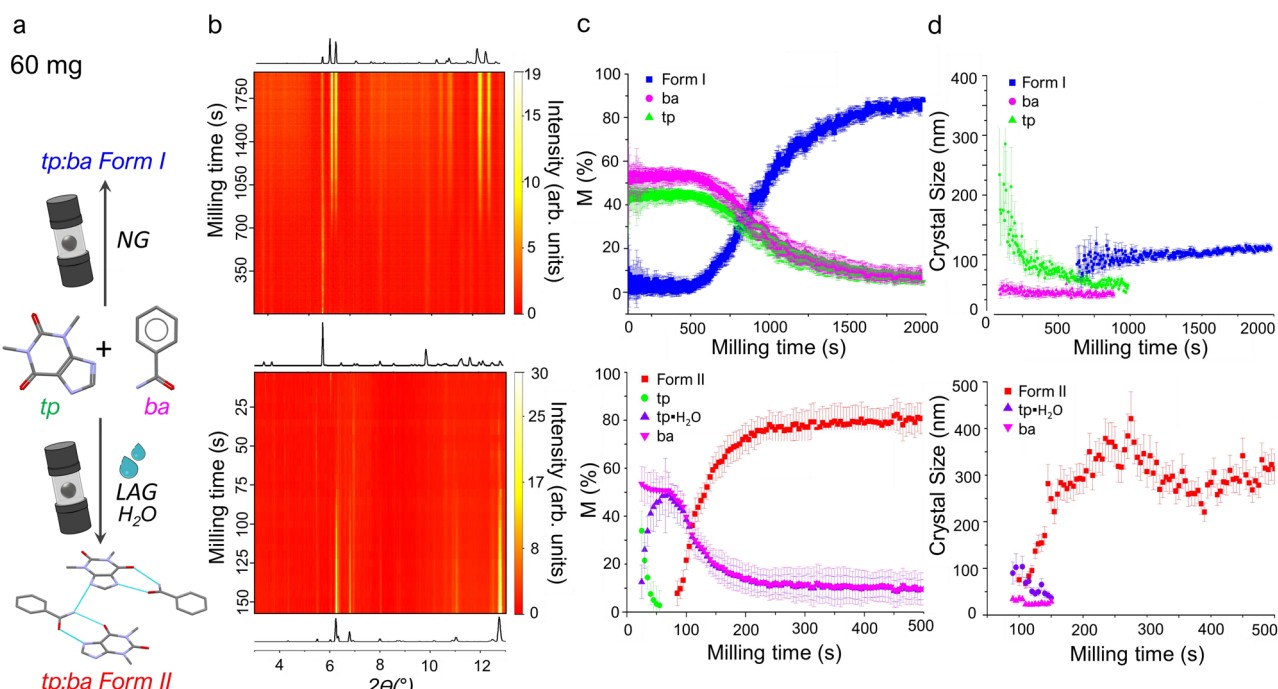

**Fig. 5 TRIS XRPD for Reaction III, the cocrystallisation of the 1:1 cocrystal of theophylline (tp) and benzamide (ba). a** Reaction III was explored under (upper, Reaction IIIa) ball mill neat grinding (NG) and (lower, Reaction IIIb) liquid-assisted grinding (LAG) with 60 mg loading and 9 µL of water. Note atom colouring as: white-hydrogen; grey-carbon; blue-nitrogen; red-oxygen. **b** Heat maps (linear intensity, arb. units) of TRIS-XRPD empirical background-subtracted data for the ball milling synthesis of (upper, Reaction IIIa) tp:ba Form I and (lower, Reaction IIIb) tp:ba Form II, alongside calculated 1D XRPD patterns for the coformer mixture (tp + ba) and the final products. **c** Quantitative phase analysis for the cocrystallisation of tp + ba obtained by Rietveld refinements, showing the consumption of (pink) ba and (green) tp, with simultaneous formation of (blue and red) the cocrystal phases with relative ESD bars. Note the formation of tp hydrate (tp·$H_2O$; purple) occurs under LAG conditions. **d** Scherrer crystal size obtained for cocrystals Form I and Form II from profile analyses of the TRIS-XRPD data with relative ESD bars (see Supplementary Notes 5.3 and 5.4 for details) are shown for phases more abundant than 20 M%.

covalent disulfide exchange Reaction IV under ball mill neat grinding (NG) conditions (Fig. 6) using our 10 mm diameter milling jar[32,42]. Literature reports two known polymorphs of the heterodimer 2NO₂-PhSSPh-4Cl, Form A and Form B, the former resulting under NG conditions.

Even at only 10 mg loading, the peak to background ratio obtained from our TRIS-XRPD approach yielded exceptionally low ESDs for sequential refinement of diffraction profile parameters (Fig. 6c). QPA suggests a short-lived induction period

(ca. 200 s) followed by the formation of the product phase, Form A (Fig. 6c). Crystallite size analysis again suggests both reactant phases reach domain sizes of ca. <100 nm prior to onset of the reaction, with the product phase equilibrating to crystallites of approximately the same size (Fig. 6d). These sizes are consistent with our previously reported crystallite sizes for Form A using conventional milling equipment[32,42]. This demonstrates unequivocally that the strategy presented here yields reliable crystal size values, even with loadings as low as 10 mg. The exchange reaction

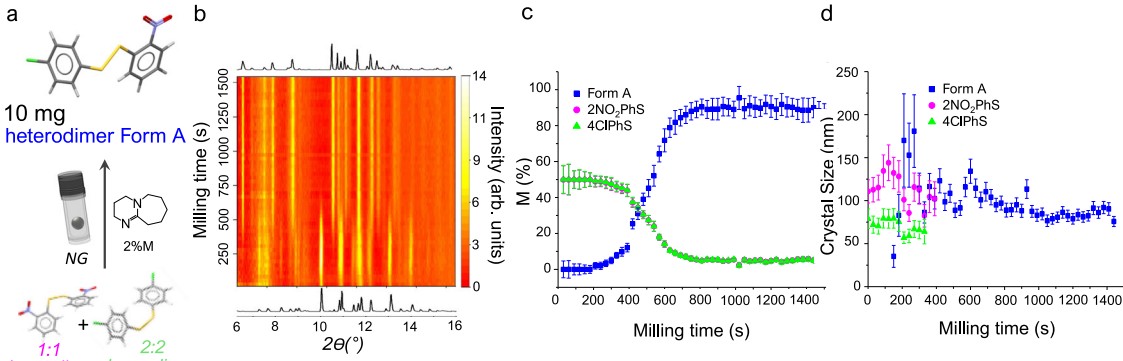

**Fig. 6 TRIS-XRPD for Reaction IV, the disulfide exchange reaction between bis(2-nitrophenyl) and bis(4-chlorophenyl) disulfide base catalysed by dbu (2 M%) with 10 mg loading under ball mill neat grinding (NG) conditions. a** The reaction scheme for the disulfide exchange reaction, with atoms coloured: white-hydrogen; grey-carbon; blue-nitrogen; red-oxygen; yellow-sulfur; green-chlorine. **b** Heat maps (linear intensity, arb. units) of TRIS-XRPD with empirical background-subtracted data for Reaction IV, alongside 1D calculated diffraction profiles for the mixture of starting reagents (homodimers) and final product phase. **c** Quantitative phase analysis of Reaction IV as obtained by Rietveld refinements, shown as molar fraction of starting materials (pink and green) and product phase (blue) with relative ESD bars. **d** Scherrer crystal size obtained from profile analysis of the TRIS XRPD data with relative ESD bars (see Supplementary Note 5.5 for details) are shown for phases more abundant than 20 M%.

goes to completion as shown by HPLC (see Supplementary Note 5.6). To the best of our knowledge, this represents the smallest-scale milling experiment with TRIS diffraction monitoring reported to date. Our methodology clearly allows for successful in situ diffraction on very small loading, which is crucial for synthetic chemistry where materials are most often scarce, expensive, and toxic.

We have demonstrated that the implementation of our strategies for data acquisition and data processing of the TRIS monitoring of ball milling reactions solves long-standing issues with the technique (Table 1), and leads to high-quality in situ diffraction data. The new miniaturised milling jars offer significant improvements in the data quality with a real representation of mechanochemical experiments under typical milling conditions. These combined strategies of data acquisition, data processing and use of very small milling jars promises exciting perspectives in terms of in situ diffraction investigations, while posing no significant experimental limit or complexity. The larger wavelength and optimised data collection strategies provide access to more defined experimental patterns. The Rietveld refinements on such experimental data, using innovative functions for peak displacement and shape, revealed exceptional details of the reactions. These include monitoring the microstructural evolution of different milling processes with excellent estimated standard deviations also for experiments on organic compounds with loading as little as 10 mg. It is now possible to explore a range of chemistries which rely on scarce, expensive, and toxic compounds, and were therefore previously inaccessible by mechanochemistry.

Our developments allow reliable real-time quantitation of mechanochemical transformations, on a par with traditional ex situ approaches. We expect this approach to become a routine strategy for the fundamental study of mechanochemistry across all areas of chemistry and materials science. Moreover, our developments pave the way to wholly new directions in X-ray-based TRIS methodology, changing the playing field of mechanochemistry and promising exciting developments in this increasingly influential field of research.

## Methods
Ball milling reactions were performed using a Fritsch Pulverisette 23 vertical vibratory ball mill. Reactions were conducted in custom-made polymethylmethacrylate (PMMA) jars (see Supplementary Note 2 for details), using 1 stainless-steel milling ball (8 mm diameter for Reactions I and II, 7 mm diameter for Reaction III and 6 mm diameter for Reaction IV), milling at a frequency of 50 Hz. A total mass loading of 70 mg of material was used for Reactions I and II, 60 mg for Reaction III and 10 mg for Reaction IV. Full details of sample preparation are provided in Supplementary Note 4. TRIS-XRPD monitoring was performed at $\mu$Spot (BESSY-II; Helmholtz-Zentrum Berlin) using a 150 $\mu$m beam monochromated (with Si(111) monochromator) to 17 keV. Scattering was collected using an Eiger 9M detector, placed ca. 250 mm from the sample. The instrumental profile was refined from a NIST Si standard, obtained from a Si powder shaking at 50 Hz. Scattering data were integrated using the DPDAK software and analysed with TOPAS V6. A detailed discussion of data collection and analysis is available in Supplementary Notes 2–6.

## Data availability
The data-processing strategies and raw data used in this work are available in the supplementary information, Source data are provided with this paper.

## Code availability
Custom-built TOPAS functions are available in the supplementary information. Complete input files for TOPAS Academic version 6[45,46] are available in the supplementary information.

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

## Acknowledgements

Experiments were conducted at μSpot (BESSY-II; Helmholtz-Zentrum Berlin). The authors thank Dr Ralf Bienert for beamline technical assistance and Ms Bettina Röder for input to the design and manufacturing of the milling jars. A.B. and P.P.M. acknowledge the COMP-HUB Initiative, funded by the "Departments of Excellence" program of the Italian Ministry for Education, University and Research (MIUR, 2018-2022). A.B., A.M.B., P.P.M. and F.E. acknowledge COST Action CA18112 – Mechanochemistry for Sustainable Industry. G.I.L. thanks the Department of Earth Sciences of the University of Cambridge for general support.

## Author contributions

A.A.L.M. and A.M.B. designed the milling jars. A.A.L.M. optimised the data collection strategy. A.A.L.M. and A.M.B. performed the milling experiments and A.A.L.M. collected the TRIS diffraction data. A.M.B. performed the HPLC ex situ analyses and G.I.L. calculated the ex situ phase composition for Reactions I and III. G.I.L. and P.P.M. coded the diffraction aberration functions and performed the Rietveld structural and microstructural analyses. A.A.L.M., G.I.L., P.P.M. and A.M.B. wrote the manuscript and ESI. G.I.L., A.A.L.M., P.P.M., A.M.B., J.K.M.S., A.B. and F.E. discussed the results and commented on the manuscript.

## Funding

## Competing interests

The authors declare no competing interests.
