## [Peer Review File. · Nature Communications]

Changing the game of time resolved X-ray diffraction on the mechanochemistry playground by downsizingEditorial Note: This manuscript has been previously reviewed at another journal that is not operating a transparent peer review scheme. This document only contains reviewer comments and rebuttal letters for versions considered at *Nature Communications*.

REVIEWERS' COMMENTS

Reviewer #2 (Remarks to the Author):

This is a resubmission of the manuscript after a major revision. I liked already the first version, but I must claim that the new version that I have received now is better. It highlights better the novelty and importance of this work as compared to what was achieved a few years ago with the participation of some of the co-authors of this manuscript. The improvements of the techniques of data collection and data processing had a synergetic effect on the type of challenging problems that can be approached now. These new problems that can be addressed using new tools suggested by the authors include, among other, the possibility to analyze reliably microstrain and particles size, to scan a sample in the milling jar in space, follow amorphization, synthesis, polymorphic transitions, double ionic exchange reactions with a precision that could not be achieved earlier. This paper will definitely have a high impact and will inspire other researchers to use the same tools in their work.

An important improvement is also that the new development of the technique gives the possibility to use lower-energy radiation for the diffraction measurements, what is important for many experiments, for example, when a powder pattern has many diffraction maxima close to each other.

I support publication of this work, and think that the level of this contribution is in fact appropriate for *Nature Communications*. At the same time, I have a few suggestions of improvement.

1. I understand, that it is not possible to give all the important references even to the most important, pivotal papers in the field of mechanochemistry that have been previously published. I can suggest that the authors in this case refer to a recent review paper (Michalchuk A.A.L. et al., *Frontiers in Chemistry*. 2021. V.9. 685789:1-29. DOI: 10.3389/fchem.2021.685789), and references therein.
2. In the section Authors Contributions I could not find information for some of the co-authors. Please, look once again and add.
3. Is it possible to justify the choice of the case studies - sample reactions - in a way more obvious from the first sight? Why not other transformations? Maybe a Scheme could help (reaction - challenge / aspect of applying the new methodology that it illustrates).
4. A Schematic comparison <What was possible / not possible to achieve with earlier described techniques and what can be done now using the newly proposed instrument and method of data treatment> would help.

This is a resubmission of the manuscript after a major revision. I liked already the first version, but I must claim that the new version that I have received now is better. It highlights better the novelty and importance of this work as compared to what was achieved a few years ago with the participation of some of the co-authors of this manuscript. The improvements of the techniques of data collection and data processing had a synergetic effect on the type of challenging problems that can be approached now. These new problems that can be addressed using new tools suggested by the authors include, among other, the possibility to analyze reliably microstrain and particles size, to scan a sample in the milling jar in space, follow amorphization, synthesis, polymorphic transitions, double ionic exchange reactions with a precision that could not be achieved earlier. This paper will definitely have a high impact and will inspire other researchers to use the same tools in their work. An important improvement is also that the new development of the technique gives the possibility to use lower-energy radiation for the diffraction measurements, what is important for many experiments, for example, when a powder pattern has many diffraction maxima close to each other.

We are grateful to the reviewer for their kind support of our manuscript, and are delighted to hear that the reviewer appreciated the novelty and importance of the developments that we have made in our article.

I support publication of this work, and think that the level of this contribution is in fact appropriate for Nature Communications. At the same time, I have a few suggestions of improvement.

- 1. I understand, that it is not possible to give all the important references even to the most important, pivotal papers in the field of mechanochemistry that have been previously published. I can suggest that the authors in this case refer to a recent review paper (Michalchuk A.A.L. et al., *Frontiers in Chemistry*. 2021. V.9. 685789:1-29. DOI: 10.3389/fchem.2021.685789), and references therein.**

Indeed, we wish there would be opportunity to cite a broader range of the seminar works done in this field, and regret we cannot. However, we have included the suggested reference, which contains many such seminal references.

- 2. In the section Authors Contributions I could not find information for some of the co-authors. Please, look once again and add**

We have now included an additional statement to incorporate the contributions of all authors, and reads: AALM, GIL, PPM, and AMB wrote the manuscript and ESI. GIL, AALM, PPM, AMB, JKMS, AB, FE discussed the results and commented on the manuscript.

- 3. Is it possible to justify the choice of the case studies - sample reactions - in a way more obvious from the first sight? Why not other transformations? Maybe a Scheme could help (reaction - challenge / aspect of applying the new methodology that it illustrates).**

Our selection of systems was in based primarily on the breadth of material types that they represent, and their popularity in the current mechanochemical literature. We explicitly selected known systems that have been previously studied by older TRIS methods, to better highlight the new insights we could obtain using our developments, most specifically for Reactions II and III. We have included an additional paragraph at the beginning of the results and discussion to better highlight this.

We elected to study archetypical examples of reactions from across the major classes of solid materials being regularly studied by ball milling:

By using reported examples, comparison against literature reports of TRIS monitoring from previous set-ups was possible (reactions I, II, and III). Moreover, our selected systems include high symmetry crystal structures (reaction I and II) with strongly scattering elements (reaction I) and poorly scattering elements (reaction II), along with low symmetry systems with poorly scattering elements (reactions III and IV). Hence, our selection of systems demonstrates unequivocally the promise and universality of our developments.

4. A Schematic comparison <What was possible / not possible to achieve with earlier described techniques and what can be done now using the newly proposed instrument and method of data treatment> would help

We thank the reviewer for this very helpful idea. We have added a table to the introduction which highlights the major challenges associated with existing TRIS methodologies, and the aspects of our developments that provide solutions to these issues.

Complexity	Proposed Solution
Sample scattering intensity and reliable XRPD refinement	 • Low energy radiation to reduce peak overlap • Minimise jar thickness to maximise sample scattering • Include experimental background (empty jar) in the PXRD data analysis by whole pattern Rietveld refinement to minimise the number of background parameters • Use experimental background scale factor to normalize PXRD scans • Sequential approach to Rietveld refinement, i.e. use the output obtained for scan number "n" as input for scan number "n + 1"
Instrumental broadening of diffraction profile	 • Low energy radiation to reduce peak overlap • Optimise beam alignment strategy to resolve multiple scattering components using a standard • Develop physically meaningful XRPD peak shape modelling for microstructural analysis • Include experimental background (empty jar) in the PXRD data analysis by whole pattern Rietveld refinement to minimise the number of background parameters • Sequential approach to Rietveld refinement • Parametric refinement for phase scale factors, i.e. constrain scale factors to sigmoidal curves
Scale of powder required for milling	 • Minimise jar thickness to maximise sample scattering • Minimise powder caking
Loss of free powder by sticking or caking on internal surfaces	 • Carefully analyse loading vs milling parameters to maximise powder distribution

5. Aside from the minor suggestions in the report below, in their confidential notes to us this Reviewer also mentioned that you could consider complementing your manuscript with a small discussion on the concerns raised by the reviewers, and how you answered them (which would be optional, in the Supplementary Information).

We have added to Supplementary Note 9 a brief discussion on the topic of our response to Reviewer 1, which reads:

When considering XRPD, a number of key features must be extractable for reliable analysis. First, accurate positions of the Bragg reflections are necessary to analyse the crystallographic unit cells under investigation. Second, accurate intensities of the reflections provide essential information regarding the atomic structure of the material, and correct phase quantification. Finally, the widths of Bragg reflections depend on the crystallinity of the material; accurate determination of these widths is therefore essential for exploring the microstructure, i.e. the crystallite size and the microstrain. Other features, such as accurate scattering background, are also significant, and can provide crucial insight into the presence of non-crystalline phases.

Typically, the quality of XRPD data derived from TRIS analysis has been suboptimal, leading to significant uncertainties regarding phase identification, crystal structure, particle size, and microstructure. Hence, robust analysis of mechanochemical transformations has been limited to *ex situ* analysis. In the present paper we have made extensive efforts to improve the data collection and processing strategies. Our thin-walled milling jars (0.5-0.7 mm) greatly reduce the background scattering, and hence enhance the scattering signal of the material within the wall. This has allowed us to reduce the quantity of material required for in situ investigation (formerly *ca.* 200-1000 mg) to only 10-60 mg. Simultaneously, the enhanced scattering from the sample increases the resolution of the diffraction profile, leading to improved confidence in the position and shape of diffraction peaks. Associated with the reduced wall thickness, our XRD data was collected at significantly lower energies than usually employed for TRIS measurements. By diffracting at 17 keV (instead of > 40 keV used at other synchrotron sources) our Bragg reflections were highly resolved, thereby avoiding any undesirable overlap of reflections, and thus providing reliable peak shapes. Finally, our data collection strategy allowed for careful alignment of the X-ray beam, thereby reducing artificial splitting of the Bragg reflections from non-ideal sample geometry. Together, these developments allowed us to construct a robust model for the structure of scattering from our set-up, and hence to extract robust and reliable data for full profile analysis. Thinner walls can of course enhance thermal conductivity. However, thermal measurements in jars with 2.5 mm walls suggested global temperature rises of only 5-10 °C over the course of a ball milling reaction.¹⁶ Thus, any change in thermal conductivity will have negligible effects on the transformation.

The loss of free flowing powder during ball milling is a critical issue for TRIS-XRPD analysis. When powder cakes, clumps, or is otherwise stuck to an internal surface of the jar, it becomes 'invisible' to the X-ray beam.¹⁵ Correspondingly, accurate analysis of reaction profiles requires such effects to be minimised. This is particularly challenging when liquid assisted grinding reactions of highly compressible materials (e.g. organic solids) is being considered. Although we cannot claim to have solved this problem in the present work, Supplementary Note 7 outlines a significant advancement towards reducing this issue. We have found that free flowing powder can be maximised by carefully controlling the fill volume. We strongly encourage all researchers involved in TRIS XRPD for ball milling reactions to conduct such preliminary work to ensure reliability of their data sets.